# Individuals with Metabolic Syndrome Show Altered Fecal Lipidomic Profiles with No Signs of Intestinal Inflammation or Increased Intestinal Permeability

**DOI:** 10.3390/metabo12050431

**Published:** 2022-05-11

**Authors:** Mia J. Coleman, Luis M. Espino, Hernan Lebensohn, Marija V. Zimkute, Negar Yaghooti, Christina L. Ling, Jessica M. Gross, Natalia Listwan, Sandra Cano, Vanessa Garcia, Debbie M. Lovato, Susan L. Tigert, Drew R. Jones, Rama R. Gullapalli, Neal E. Rakov, Euriko G. Torrazza Perez, Eliseo F. Castillo

**Affiliations:** 1University of New Mexico School of Medicine, University of New Mexico Health Sciences Center, Albuquerque, NM 87131, USA; mjcoleman@salud.unm.edu (M.J.C.); lmespino@salud.unm.edu (L.M.E.); helebensohn@salud.unm.edu (H.L.); 2Clinical and Translational Science Center, University of New Mexico Health Sciences Center, Albuquerque, NM 87131, USA; zimkutem@gmail.com (M.V.Z.); jmgross@salud.unm.edu (J.M.G.); nlistwan@unm.edu (N.L.); sacano@salud.unm.edu (S.C.); vygarcia@salud.unm.edu (V.G.); dlovato@salud.unm.edu (D.M.L.); stigert@salud.unm.edu (S.L.T.); 3Division of Gastroenterology and Hepatology, Department of Internal Medicine, University of New Mexico Health Sciences Center, Albuquerque, NM 87131, USA; nyaghooti@gmail.com (N.Y.); christinaling837@gmail.com (C.L.L.); nrakov@salud.unm.edu (N.E.R.); etorrazzaperez@salud.unm.edu (E.G.T.P.); 4Metabolomics Core Resource Laboratory, New York University Langone Health, New York, NY 10016, USA; drew.jones@nyulangone.org; 5Department of Pathology, University of New Mexico Health Sciences Center, Albuquerque, NM 87131, USA; rgullapalli@salud.unm.edu

**Keywords:** metabolic syndrome, metabolomics, lipidomics, dyslipidemia

## Abstract

Background: Metabolic Syndrome (MetS) is a clinical diagnosis where patients exhibit three out of the five risk factors: hypertriglyceridemia, low high-density lipoprotein (HDL) cholesterol, hyperglycemia, elevated blood pressure, or increased abdominal obesity. MetS arises due to dysregulated metabolic pathways that culminate with insulin resistance and put individuals at risk to develop various comorbidities with far-reaching medical consequences such as non-alcoholic fatty liver disease (NAFLD) and cardiovascular disease. As it stands, the exact pathogenesis of MetS as well as the involvement of the gastrointestinal tract in MetS is not fully understood. Our study aimed to evaluate intestinal health in human subjects with MetS. Methods: We examined MetS risk factors in individuals through body measurements and clinical and biochemical blood analysis. To evaluate intestinal health, gut inflammation was measured by fecal calprotectin, intestinal permeability through the lactulose-mannitol test, and utilized fecal metabolomics to examine alterations in the host–microbiota gut metabolism. Results: No signs of intestinal inflammation or increased intestinal permeability were observed in the MetS group compared to our control group. However, we found a significant increase in 417 lipid features of the gut lipidome in our MetS cohort. An identified fecal lipid, diacyl-glycerophosphocholine, showed a strong correlation with several MetS risk factors. Although our MetS cohort showed no signs of intestinal inflammation, they presented with increased levels of serum TNFα that also correlated with increasing triglyceride and fecal diacyl-glycerophosphocholine levels and decreasing HDL cholesterol levels. Conclusion: Taken together, our main results show that MetS subjects showed major alterations in fecal lipid profiles suggesting alterations in the intestinal host–microbiota metabolism that may arise before concrete signs of gut inflammation or intestinal permeability become apparent. Lastly, we posit that fecal metabolomics could serve as a non-invasive, accurate screening method for both MetS and NAFLD.

## 1. Introduction

The global incidence of Metabolic Syndrome (MetS), affecting over 25% of the global population (~1.97 billion) and 33% of those living in the United States, has severe health and economic consequences [1,2,3]. MetS is comprised of multiple dysregulated metabolic pathways that can cause or result in insulin resistance [4]. Current diagnostic criteria for MetS must include three out of the five risk factors: hypertriglyceridemia, low high-density lipoprotein (HDL) cholesterol, hyperglycemia, elevated blood pressure, or increased abdominal obesity [5]. MetS is useful in detecting patients at high risk for other metabolic diseases including cardiovascular disease (CVD) [6,7], type 2 diabetes (T2D) [8,9], and even hepatocellular carcinoma (HCC) [10,11].

The liver plays a central role in the pathogenesis of MetS. Glucose and triglycerides (TG) are produced in the liver. When the liver is insulin resistant, the “brakes” on glucose and TG production are lost [12,13]. Hypertriglyceridemia, high levels of TG, causes hepatic fat accumulation and organ dysfunction, further contributing to hepatic insulin resistance [14,15,16]. Excessive fat in the liver unrelated to alcohol use, viral infections, or drugs has been termed Non-Alcoholic Fatty Liver Disease (NAFLD) [17,18,19]. Similar to MetS, over a billion people worldwide are affected by NAFLD [20]. NAFLD is also increasingly diagnosed in children [21]. This is alarming given that the trajectory of the disease burden in children can be decades longer than patients who develop NAFLD later in life. In the United States, health care costs directly related to NAFLD are estimated to be USD 100 billion annually [21]. NAFLD provides a pathophysiological “timeline” of hepatic pathology. This begins with fat accumulation (steatosis), fat accumulation with inflammation (non-alcoholic steatohepatitis, NASH) and the possibility of subsequent progression to liver cirrhosis and HCC [10,22]. However, unlike MetS, NAFLD has specific histopathologic markers. Steatosis is defined as 5–10% of fatty hepatocytes, and steatohepatitis often exhibits ballooning necrosis, inflammation, and fibrosis [23]. Although NAFLD is clinically less ambiguous to diagnose than metabolic syndrome, a biopsy is currently required to diagnose NAFLD and NASH. MetS is defined in many ways by various organizations, making it somewhat amorphous. Nevertheless, due to their closely overlapping mechanisms, NAFLD and MetS can initiate each other and predict the same disease likelihood in high-risk patients [24,25,26,27]. While not all patients inevitably acquire comorbid metabolic derangements, cirrhosis, or malignancy, many do, warranting early intervention and clear diagnostic criteria.

Evidence suggests the gastrointestinal (GI) tract may play a significant role in metabolic diseases [28,29,30,31,32]. There is a tripartite interaction in the GI tract in which the gut microbiota, the immune system, and the intestinal epithelium maintain the balance between intestinal homeostasis and inflammation [33,34]. Dysfunction in one of these components can have profound effects on the other two systems and contributes to metabolic dysfunction [28,35]. Interestingly, gut dysbiosis and increased intestinal permeability have been observed in individuals with NAFLD and in animal models of NAFLD, suggesting a role for the GI tract in the etiology of NAFLD [36,37,38,39,40,41,42,43,44,45,46,47,48,49,50,51,52,53,54,55,56]. Dysbiosis is also associated with obesity and T2D morbidity and disease course, influencing inflammation, gut permeability, immune function, insulin resistance, and lipid metabolism [57,58,59,60,61,62]. Given the reciprocal gut–liver interaction, examining the GI tract in MetS patients could prove beneficial in both interventional strategies and preventative diagnostics.

The goal of our pilot study was to determine if human subjects with MetS have intestinal inflammation and increased intestinal permeability similar to other metabolic diseases. Additionally, we sought to examine fecal metabolites associated with our clinical phenotype and to further understand metabolic variation as well as interactions between the gut microbiota–host. Our data indicated there was a noticeable difference in fecal lipidomics and some had a strong correlation with both increasing triglyceride and fasting insulin levels. However, there was not a significant difference in intestinal permeability or inflammation between MetS subjects and controls, suggesting metabolic perturbations may arise before gut inflammation and intestinal permeability.

## 2. Results

### 2.1. Goal of the Study

In this pilot study, we sought to understand differences in gut health in individuals with metabolic syndrome (MetS) compared to non-metabolic syndrome (control) participants. Specifically, examining differences in intestinal inflammation, intestinal permeability, and fecal metabolites as an insight between diet–microbiota–host interactions. This pilot study was approved by UNM HSC HRRC (see Section 6) and participants were recruited from and seen at the UNM CTSC clinic in a two-week period. Participants were classified as having MetS or normal based on the established criteria described in the Section 6.

### 2.2. Clinical and Biochemical Analysis of Study Cohort

The study population consisted of 18 individuals who were seen under fasting conditions. The demographics of the study cohort are shown in Table 1. After body measurements, vital signs, and blood sample analyses were taken, 10 participants were classified as controls and 8 participants as MetS. Assessment of the MetS risk factors revealed the MetS group had increased abdominal obesity (Figure 1A) and showed signs of dyslipidemia as the triglycerides were significantly higher (Figure 1B and Appendix A) and HDL cholesterol was significantly lower (Figure 1C and Appendix A). Body measurements revealed a significant increase in body weight as well as body mass index (BMI) in our MetS cohort with no difference in the waist-to-hip ratio or height (Appendix A). Bioelectrical impedance analysis further revealed the MetS group had a significant increase in the percent of body fat as well as a higher fat mass with no difference in lean mass (Appendix A). Examination of fasting glucose levels revealed no difference between both groups (Figure 1D); however, both fasting insulin levels (Figure 1E) and Hemoglobin A1C (HbA1C) levels (Figure 1F) were significantly higher in the MetS group. Calculation of Homeostatic Model Assessment for Insulin Resistance (HOMA-IR), an indicator of insulin resistance, revealed the MetS group had a higher HOMA-IR score (Appendix A). The calculation of insulin sensitivity via quantitative insulin sensitivity check index (QUICKI) revealed the MetS group had a lower insulin sensitivity score (Appendix A) [63,64,65]. Together the HOMA-IR and QUICKI scores suggested the MetS group showed signs of insulin resistance. Blood pressure and mean arterial pressure (MAP) trended higher in the MetS group but were not significantly different to that of controls (Figure 1G–I). Lastly, we found no significant differences in the comprehensive metabolic panel (CMP) between groups including aspartate transaminase (AST), alanine aminotransferase (ALT), or AST/ALT ratios (Appendix A). Collectively, our MetS cohort showed significant differences in abdominal obesity, dyslipidemia, and insulin resistance.

### 2.3. Metabolic Syndrome Participants Showed Systemic Inflammation That Correlated with Dyslipidemia

Metabolic disorders are frequently associated with low-grade inflammation [66]. The term metabolic inflammation characterizes a low-level of systemic inflammation. As such, several studies have associated these conditions with increased circulating levels of acute phase proteins and cytokines such as C-reactive protein (CRP) and TNFα, respectively. To determine the level of metabolic inflammation occurring in our two groups, we examined the serum levels of both TNFα and high-sensitivity C-reactive protein (hsCRP) as proxies for metabolic inflammation [67,68]. Serum TNFα levels were found to be significantly higher in the MetS group (Figure 2A). HsCRP levels were slightly higher in the MetS group; however, this difference was not significant (Figure 2B). Intriguingly, there was a strong positive correlation between increasing TNFα levels and increasing triglyceride levels (r = 0.7978; *p* = 0.0177) (Figure 2D). Rising TNFα levels also had a strong negative correlation with decreasing HDL cholesterol levels (r = −0.7094; *p* = 0.0488) (Figure 2E). These results are consistent with the previously reported correlation between TNFα levels and dyslipidemia [69,70].

More recently, attention has been drawn to the GI tract as a possible etiological factor driving metabolic disorders [29,30,31,32,35,59,62,66,71]. In fact, MetS and NAFLD are frequently reported in patients with inflammatory bowel disease (IBD) [72,73,74,75,76]. Therefore, we examined the level of intestinal inflammation through a fecal calprotectin test. This noninvasive test provides a functional quantitative measure of intestinal inflammation [77,78,79]. Interestingly, we saw no difference in fecal calprotectin levels between the control and MetS groups (Figure 2C). Further analysis of the GI tract revealed no significant difference in intestinal barrier permeability as the lactulose to mannitol ratio was similar between both groups (Figure 2F), as were the overall levels of recovered urine lactulose (Figure 2G) and mannitol (Figure 2H). This test allows for the quantification of two non-metabolized sugar molecules (i.e., lactulose and mannitol) to permeate the intestinal barrier [80]. Taken together, our data suggest the MetS group had a low-level of systemic inflammation but no observable signs of intestinal inflammation or barrier dysfunction.

### 2.4. Metabolomics Revealed Altered Fecal Metabolites in Metabolic Syndrome Participants

Utilizing untargeted lipidomic analyses [81,82], we sought to identify the lipids associated with our clinical phenotype. Specifically, we analyzed fecal samples from our control and MetS groups to further characterize the GI tract. Figure 3A shows a volcano plot of all 7453 lipid features detected. The red dots on the right represent lipids with higher levels in MetS participants, while the red dots on the left are lipids with lower levels in MetS participants. The MetS group had 417 lipid features that were significantly different from control participants (Figure 3A). The putative identification derived from LIPID MAPS^®^ Structure Database (LMSD) [83] utilizing observed *m*/*z* was determined for the top 20 lipid features that showed the highest fold change in MetS fecal samples (Table 2). Out of these 20 lipids, LMSD predicted they were glycerolipids, glycerophospholipids, sphingolipids, fatty acyls, and polyketides (Figure 3B). For brevity, we also show the 30 with the lowest *p*-values in Appendix A. Among these 30 lipids, LMSD predicted that all were still glycerolipids (*n* = 9), glycerophospholipids (*n* = 18), and sphingolipids (*n* = 3) (Appendix A). The lipid feature that was most significantly decreased in MetS fecal samples could not be identified by LMSD. Fecal samples were also assessed for approximately 150 polar metabolites that cover much of the central carbon metabolism pathways. The principal components analysis (PCA) plot and heatmap of metabolites revealed no overall clustering of control or MetS group-derived fecal metabolites (Appendix A). However, the volcano plot revealed two metabolites which were significantly different between groups using the a priori cutoffs of [log_2_FC] ≥ 2, *p* < 0.05 (Figure 4A). Orotic acid was significantly higher in MetS participants, while the left side shows that Carnosine was significantly lower in MetS participants (Figure 4A) in MetS fecal samples (Figure 4B,C). Interestingly, carnosine is a dipeptide of βalanine and histidine, and is a normal product of the liver, while orotic acid is a key intermediate in de novo pyrimidine nucleotide synthesis (HMDB 5.0) [84]. Intriguingly, five fecal lipids that belong to the glycerolipid, glycerophospholipids, and sphingolipids categories showed a strong positive correlation with triglyceride and fasting insulin levels (Appendix A; statistics shown in Appendix A). PC 12:0_20:4 (Diacyl-glycerophosphocholine, PCaa), a glycerophosphocholine, showed a strong positive correlation with increasing triglycerides (r = 0.66; *p* = 0.0041), serum TNFα (r = 0.50; *p* = 0.0424), and fasting insulin levels (r = 0.71; *p* = 0.0015) as well as strong negative correlation with decreasing HDL cholesterol levels (r= −0.54; *p* = 0.0267) (Appendix A). Given fecal metabolites can provide insight into host–microbiota–diet interactions, our data suggest major alterations in the intestinal metabolism, in the absence of localized intestinal inflammation, in MetS subjects. Lastly, our data reveal that fecal lipids could provide an insight into clinical phenotypes and could serve as an alternative noninvasive method to diagnose MetS and possibly NAFLD.

## 3. Discussion

In this present study, we evaluated intestinal homeostasis in individuals with or without MetS. Interestingly, MetS participants showed no signs of intestinal inflammation or increased intestinal permeability when compared to our control group. Nonetheless, we found major differences in the gut lipidome, specifically, an increase in various types of glycerolipids, glycerophospholipids, sphingolipids, fatty acyls, and polyketides, in our MetS cohort. One fecal lipid that was identified, a diacyl-glycerophosphocholine, was increased in our MetS cohort and showed a strong correlation with several MetS risk factors. Furthermore, we found our MetS cohort had a low-level of circulating TNFα that also correlated with increasing triglyceride and fecal diacyl-glycerophosphocholine levels as well as decreasing “good” HDL cholesterol levels. Taken together, our main results show that MetS subjects showed major alterations in intestinal lipid profiles suggesting alterations in the intestinal host and microbiota metabolism which may precede intestinal dysfunction.

MetS and NAFLD can both predict similar diseases including T2D, CVD, and NASH [9,26,27,85,86]. In addition, the liver is a shared focal point for both metabolic disorders as glucose and triglycerides are overproduced in the liver. The increase in triglycerides can lead to fat accumulation and is often associated with hepatic insulin resistance [13,14,15,16]. Unfortunately, both metabolic disorders can go undiagnosed as the individual can appear asymptomatic. Given the role of the liver in these two metabolic diseases, liver enzymes (e.g., ALT, AST, ALT:AST) could provide clues in relation to disease progression. However, we observed no differences in these liver enzymes between our study groups (Appendix A). Liver enzymes are also often normal in NAFLD patients and therefore are not consistent diagnostic markers [14]. The gold standard of NAFLD diagnosis relies on a liver biopsy. Liver biopsy is an invasive procedure with many absolute contraindications (coagulopathies, recent NSAID use, inability to identify an appropriate biopsy site) and relative contraindications (morbid obesity, infection, ascites). A liver biopsy is also handicapped by only being able to capture pathology in a specific moment in time. NAFLD is a chronic inflammatory disease. Like many chronic inflammatory disorders, NAFLD can have a dynamic relapsing–remitting pattern [66]. Over a short period of time NAFLD can oscillate between steatosis and steatohepatitis [66]. Fibrosis can flare and spontaneously regress [66]. Additionally, a liver biopsy cannot accurately assess a fluctuating disease process. It is therefore not appropriate to perform liver biopsies on all patients with suspected NAFLD or MetS, even if a biopsy is medically feasible [87]. Safer, faster, and more accessible testing is needed. Metabolomics may offer a non-invasive, accurate method of screening for both MetS and NAFLD. Metabolomics can analyze and quantify metabolites and lipids linked to metabolic pathways and changes could offer insight into clinical phenotypes [88,89,90,91,92,93].

Mining biofluids such as plasma, serum, urine, and even stool scan help identify biomarkers for diseases. Recently, metabolomic signatures were identified for individuals with MetS using plasma [93] and urine [91] biofluids which ultimately provided insight into MetS occurrence and progression. Unlike other biofluids, a stool can also give a comprehensive look into the GI tract as it contains microorganisms, microbial by-products, nutrients such as fibers and lipids, and inflammatory molecules. Thus, stool samples can provide molecular clues into GI health. For instance, bacterial fermentation of dietary fiber can generate metabolites such as short chain fatty acids (SCFAs) such as butyrate, propionate, and acetate that in turn modulate microbiota composition, intestinal epithelial and immune cell function, and lipid metabolism [94,95,96,97,98,99,100,101]. When the production of SCFAs is decreased from dysbiosis, it can subsequently derail the barrier and immune functions as well as the lipid metabolic pathways. Our metabolomic analyses of stool samples revealed major alterations in the gut lipidome in individuals with MetS. We observed increases in glycerophospholipids such as glycerophosphocholines as well as ceramides, a type of sphingolipids. Both glycerophosphocholines and ceramides are increased in the serum of NAFLD and NASH patients [102,103,104]. They are also strongly associated with CVD and T2D [105,106,107,108,109]. A reduction in ceramides can improve hepatic steatosis and insulin sensitivity [110,111]. Interestingly, gut microbiota-produced sphingolipids can be taken up by the intestine [112] and can enter into host metabolic pathways increasing hepatic ceramide levels [113]. In addition to changes in fecal lipids, our MetS cohort also showed an increase in orotic acid, an intermediate of pyrimidine nucleotide biosynthesis, in stool samples. Similar to the lipids described above, orotic acid has also been linked to metabolic risk factors such as hypertension [114] and can induce NAFLD in a various rodent models [115,116]. Carnosine, which was decreased in our MetS group, has proven beneficial in reducing abdominal obesity, blood pressure, and glucose in humans and animal models [117,118,119,120,121]. Overall, our observation of differential lipids and metabolites that are associated with clinical phenotypes suggest stool samples could prove beneficial as a diagnostic or preventative biofluid for metabolic disorders.

## 4. Conclusions

Our goal in this pilot study was to examine GI health in individuals with MetS. This cohort showed no signs of intestinal inflammation or increase in intestinal permeability. Animal models utilizing high-fat diets (plus glucose) to induce obesity, metabolic endotoxemia, and insulin resistances show alterations in the gut microbiota [28,35]. In addition, these models have been instrumental in showing that high-fat diets also cause an increase in intestinal permeability and inflammation [30,59,122,123,124]. In human subjects, intestinal inflammation has been observed in more advanced liver diseases such as cirrhosis and HCC [125,126]. IBD patients also can develop MetS and NAFLD while NAFLD and NASH patients have an increased risk of developing CRC [72,73,74,75,76,127,128,129]. Targeting the GI tract with probiotics in NAFLD and NASH patients has proved beneficial in reducing liver enzymes, hepatic inflammation, hepatic steatosis, and hepatic fibrosis, further supporting a role for the GI tract [130,131,132,133,134,135,136,137,138,139]. Nevertheless, these studies still do not completely explain the cause of gut dysbiosis and decreased barrier function, the increased risk of IBD and CRC in NAFLD patients, or how the probiotics are working. Thus, there is a critical gap in knowledge regarding how the GI tract, possibly through host–microbiota metabolic interaction, is involved in metabolic diseases. We posit that our MetS cohort showed no signs of intestinal dysfunction because changes in the host–microbiota metabolism precede inflammation [140]. Future endeavors to characterize gut metabolism could provide an insight into the etiology of metabolic disorders such as MetS and NAFLD.

## 5. Limitation of the Study

A major strength of this study was the examination and comparison of human subjects with or without MetS. We were able to identify changes in fecal lipidomics in our MetS cohort that had a strong correlation with several MetS risk factors. Further and contrary to animal studies, we found that individuals with MetS showed no signs of intestinal inflammation or increased permeability. Finally, our study cohort was both gender and ethnically diverse. Nevertheless, we recognize our pilot study had several limitations. These included our relatively small sample size for both populations (*n* = 10 controls and *n* = 8 MetS) that may not be truly representative of the U.S. population. Second, our volunteered “healthy” cohort in our pilot study had a few subjects with elevated blood pressure (2/10 of subjects) and high triglyceride levels (1/10; but did not have elevated blood pressure or low HDL levels). However, our control cohort did not meet the guidelines required to be diagnosed with MetS. Lastly, we believe our study could benefit from the examination of colonic biopsies from both cohorts to compare metabolic and inflammatory pathways in the colonic epithelium. This could provide us with a better understanding of the host–microbiota interactions occurring in the colon of MetS subjects and how these pathways can contribute to metabolic dysfunction.

## 6. Methods

### 6.1. Participants

Inclusion criteria for MetS participants consisted of individuals between the ages of 30–60 years with at least three of the five risk factors of MetS. The risk factors included (i) abdominal obesity: waist circumference ≥ 102 cm in men or ≥88 cm in women; (ii) elevated triglycerides: ≥150 mg/dL, or drug treatment for high triglycerides; (iii) low HDL-Cholesterol: <40 mg/dL in men or <50 mg/dL in women, or drug treatment for low HDL-Cholesterol; (iv) elevated blood pressure: systolic ≥ 130 mm Hg and/or diastolic ≥ 85 mm Hg, or drug treatment for hypertension; and (v) elevated fasting plasma glucose: ≥100 mg/dL, or drug treatment for elevated glucose. Inclusion criteria for the control group consisted of individuals aged 30–60 years that did not have MetS. Exclusion criteria for both groups included individuals who had been previously diagnosed with inflammatory bowel disease, diabetes, severe hepatic dysfunction, pregnant females, lactating/breastfeeding individuals, currently on nonsteroidal anti-inflammatory drugs (NSAIDs), protein pump inhibitors, ongoing alcohol or substance abuse via AUDIT [141] questionnaire screening to determine whether the participant’s behaviors were suggestive of alcohol abuse. Widely used in clinical settings, AUDIT screens an individual based on alcohol intake, alcohol dependence, and alcohol-related harm by formulating an overall score, with each question providing a score from 0 to 4. Lastly, individuals with the inability to render informed consent were also excluded from the study.

### 6.2. Clinical Visit

Consented participants were instructed to visit the Clinical and Translational Science Center (CTSC) clinic after an overnight fast or a minimum of 8 h of fasting. Blood was drawn to determine fasting glucose and insulin levels, high-sensitivity C-reactive protein (hs-CRP) levels, comprehensive metabolic panel (CMP), and lipid (triglycerides, total cholesterol, HDL, and LDL cholesterol) profiles (TriCore Reference Laboratories, Albuquerque, NM, USA). Additionally, Hemoglobin A1C (HbA1C) (Siemens DCA System, Singapore, Singapore) and tumor necrosis factor-alpha (TNFα) (R&D Systems, Minneapolis, MN, USA) were also analyzed (CTSC). Participants’ height, weight, waist, waist-to-hip ratio, and body composition via bioelectrical impedance were recorded. Participants were instructed to collect 10 g of stool for metabolomics (PRECISION™ Stool Collection System, Covidien, Dublin, Ireland) and fecal calprotectin. For the calprotectin assay, the stool was collected in a Calprotectin ELISA Stool Sample Collection Kit and was run on the corresponding ELISA kit (Eagle BioSciences, Inc., Amherst, NH, USA).

### 6.3. Intestinal Permeability Assay

Within two weeks after the initial visit, participants visited the CTSC clinic after fasting overnight and provided a pre-test urine sample. Participants then ingested 50 mL of solution containing 5 g of lactulose and 2 g of d-mannitol followed immediately by 200 mL of water. After 3 h, participants provided a post-test urine sample. The levels of lactulose, d-mannitol, and lactulose-mannitol ratios were assessed in the urine via ELISA (Megazyme F-FRUGL, Megazyme E-MNHPF, Bray, Ireland) [142,143,144].

### 6.4. Fecal Metabolomics

The collected 10 g of stool (PRECISION™ Stool Collection System, Covidien, Dublin, Ireland) were sent to NYU Langone Metabolomics Core Resource Laboratory to examine fecal metabolites and lipids. Hybrid metabolomics was performed examining a standard panel of ~150 polar metabolites covering much of the central carbon metabolism, and other common metabolites of interest. Separation and identification were carried out with HILIC chromatography and a library of *m*/*z* and retention times adapted from the Whitehead Institute [145], and verified with authentic standards and/or high resolution MS/MS spectral manually curated against the NIST14MS/MS and METLIN (2017) tandem mass spectral libraries [145,146].

Global lipidomics analyses were performed to profile changes in polar lipids in a data-dependent fashion. Samples were subjected to an LCMS analysis to detect and identify phospholipid molecules and quantify the relative levels of identified lipids. A lipid extraction was carried out on each sample based on published methods [81,82]. The dried samples were resolubilized in 10 μL of a 4:3:1 mixture (isopropanol:acetonitrile:water) and analyzed by UPLC-MS/MS with a modified polarity switching method [81,82]. The LC column was a Waters^TM^ CSH-C18 (2.1 × 100 mm, 1.7 μm) coupled to a Dionex Ultimate 3000^TM^ system (Dionex, Sunnyvale, CA, USA) and the column oven temperature was set to 55 °C for the gradient elution. The flow rate of 0.3 mL/min was used with the following buffers; (A) 60:40 acetonitrile:water, 10 mM ammonium formate, 0.1% formic acid and (B) 90:10 isopropanol:acetonitrile, 10 mM ammonium formate, 0.1% formic acid. The gradient profile was as follows: 40–43% B (0–1.25 min), 43–50% B (1.25–2 min), 50–54% B (2–11 min), 54–70% B (11–12 min), 70–99% B (12–18 min), 70–99% B (18–32 min), 99–40% B (23–24 min), hold 40% B (1 min). Injection volume was set to 1 μL for all analyses (25 min total run time per injection). MS analyses were carried out by coupling the LC system to a Thermo Q Exactive HF^TM^ mass spectrometer operating in heated electrospray ionization mode (HESI). Method duration was 20 min with a polarity switching data-dependent Top 10 method for both positive and negative modes. Spray voltage for both positive and negative modes was 3.5 kV, and the capillary temperature was set to 320 °C with a sheath gas rate of 35, aux gas of 10, and max spray current of 100 μA. The full MS scan for both polarities utilized a 120,000 resolution with an AGC target of 3 × 10^6^ and a maximum IT of 100 ms, and the scan range was from 350 to 2000 *m*/*z*. Tandem MS spectra for both the positive and negative modes used a resolution of 15,000, an AGC target of 1 × 10^5^, a maximum IT of 50 ms, an isolation window of 0.4 *m*/*z*, an isolation offset of 0.1 *m*/*z*, a fixed first mass of 50 *m*/*z*, and 3-way multiplexed normalized collision energies (nCE) of 10, 35, and 80. The minimum AGC target was 5 × 10^4^ with an intensity threshold of 1 × 10^6^. All data were acquired in profile mode. The top scoring structure match for each data-dependent spectrum was returned using an in-house script for MSPepSearch_×64 against the LipidBlast tandem mass spectral library of lipids [147]. Putative lipids were sorted from high to low by their reverse dot scores, and duplicate structures were discarded, retaining only the top-scoring MS2 spectrum and the neutral chemical formula, detected *m*/*z*, and detected polarity (+ or −) of the putative lipid was recorded. The resulting lipids were further identified manually by searching the accurate mass data against the LIPID MAPS^®^ Structure Database (LMSD) utilizing the observed *m*/*z* [83].

### 6.5. Statistical Analysis

Statistical analysis was performed as described in figure legends and the plots generated were obtained using the Prism software. Shapiro–Wilk tests were performed to determine whether the outcome variables were normally distributed. Two-tailed unpaired Student’s *t*-tests were used for variables that passed the Shapiro–Wilk test for normality (i.e., *p* > 0.05), and two-tailed Mann–Whitney U tests were used for variables that were not normally distributed (i.e., Shapiro–Wilk *p* < 0.05) was performed. Plots display the median (±minimum and maximum) or mean (±SE). Pearson’s Correlation Coefficients were acquired using Prism software. Fecal metabolomics data were processed as described above and analyzed by NYU Langone Metabolomics Core Resource Laboratory using their in-house analysis pipeline. Cluster analysis was performed using heatmap3 [148] package in R. Raw *p*-values < 0.05 were used as a significance threshold for prioritizing hits of interest. Principle component analysis was conducted in Python using the Scikit-learn, matplotlib, Numpy, and Scipy [149,150,151,152]. All other data were analyzed using the two-tailed unpaired Student’s *t*-test (Prism).

## Figures and Tables

**Figure 1 metabolites-12-00431-f001:**
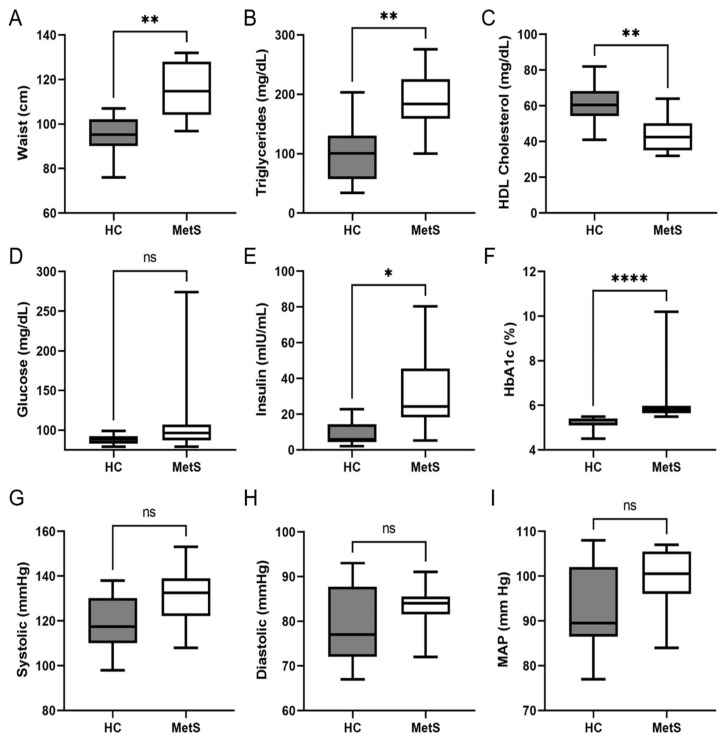
**Metabolic Syndrome risk factors.** Clinical and biochemical analysis of healthy controls (HC) and metabolic syndrome (MetS) participants. Graph showing (**A**) abdominal obesity (i.e., waist circumference); (**B**) triglyceride levels; (**C**) HDL Cholesterol; (**D**) fasting glucose levels; (**E**) fasting insulin levels; (**F**) hemoglobin A1c (HbA1c) levels; (**G**) systolic blood pressure; (**H**) diastolic blood pressure; and (**I**) mean Arterial Pressure (MAP). Graphs indicate median (±minimum and maximum). * *p* < 0.05, ** *p* < 0.01, **** *p* < 0.0005 and ns, not significant. Two-tailed unpaired Student’s *t*-tests (**A**–**C**,**G**–**I**) or two-tailed Mann–Whitney U (**D**–**F**).

**Figure 2 metabolites-12-00431-f002:**
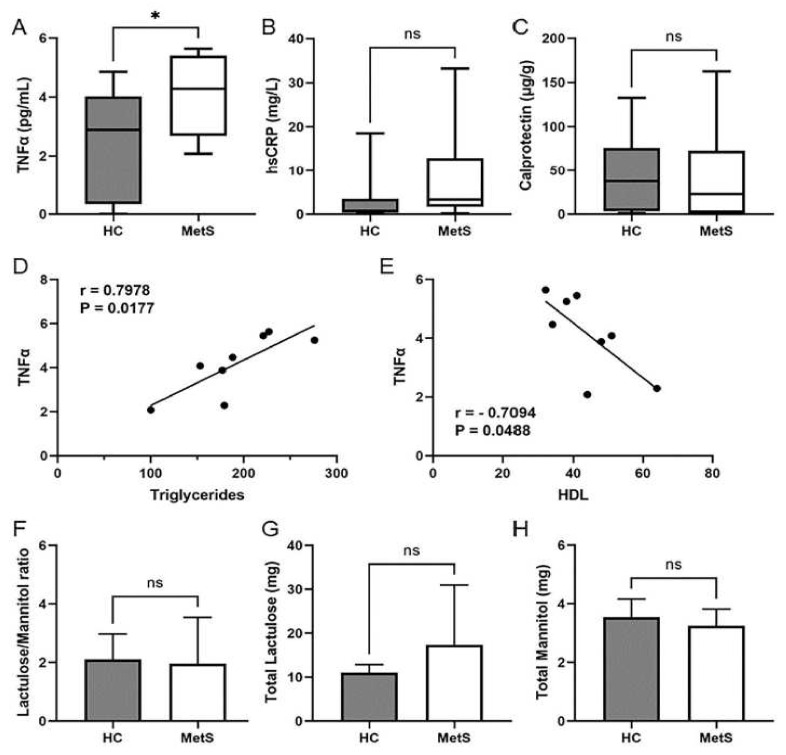
**Assessment of systemic and intestinal inflammatory markers**. Serum and fecal levels of inflammatory markers were measured in HC and MetS participants. Plot showing (**A**) serum TNFα levels; (**B**) serum hsCRP levels; (**C**) fecal calprotectin levels. (**D**,**E**) Pearson’s correlation coefficients between (**D**) TNFα and triglycerides and (**E**) TNFα and HDL cholesterol. Plot showing (**F**) lactulose/mannitol ratio; (**G**) total lactulose levels recovered in the urine; and (**H**) total mannitol levels recovered in the urine. Plots indicate median (±minimum and maximum) or mean (±SE). * *p* < 0.05, and ns, not significant. Two-tailed unpaired Student’s *t*-tests (**A**,**H**) or two-tailed Mann–Whitney U (**B**,**C**,**F**,**G**).

**Figure 3 metabolites-12-00431-f003:**
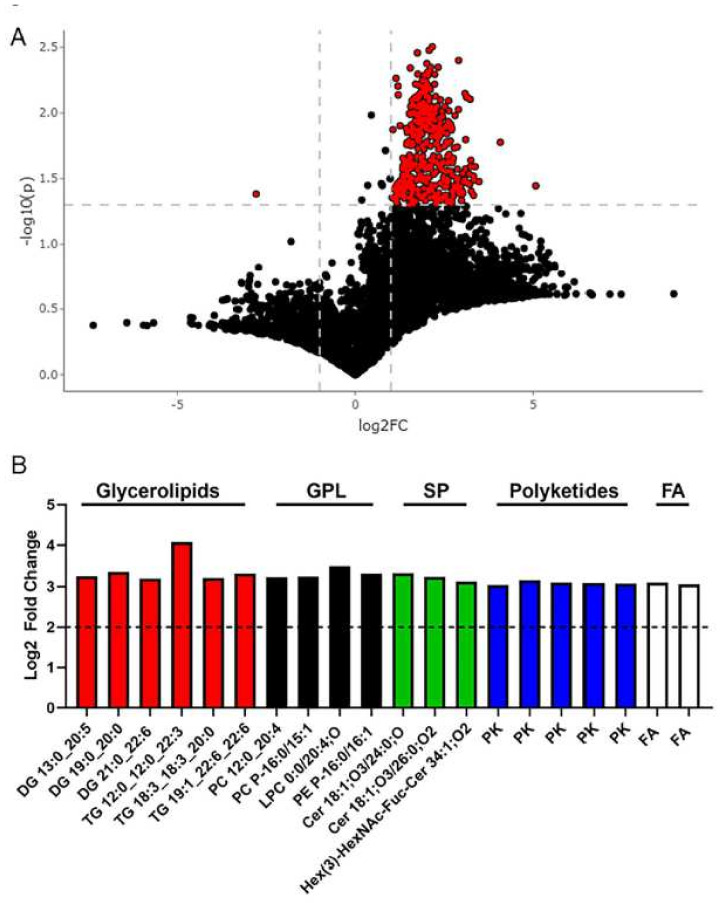
**Untargeted lipidomics show major fecal lipid variations.** (**A**) Volcano plot from UPLC-MS/MS-based untargeted lipidomics of stool from MetS and HC subjects (*n* = 7–10/group) depicting the 7453 lipids features obtained following MS data processing. Metabolite peak intensities were extracted according to a library of *m*/*z* values and retention times developed with authentic standards. Intensities were extracted with an in-house script with a 10-ppm tolerance for the theoretical *m*/*z* of each metabolite, and a maximum 30 s retention time window. Each dot represents one lipid, dashed lines indicate default thresholds for significance (*p* < 0.05) and fold change up- or down-regulation by 2-fold (Log2FC = 1). The red dots on the right represent the lipids with higher levels in MetS participants, while the dots on the left are the lipids with lower levels in MetS with respect to HCs. (**B**) Plot showing the top 20 LMSD identified lipids with highest fold change (mean; *p* < 0.05). GPL, glycerophosholipids; SP, sphingolipids; FA, fatty acyls.

**Figure 4 metabolites-12-00431-f004:**
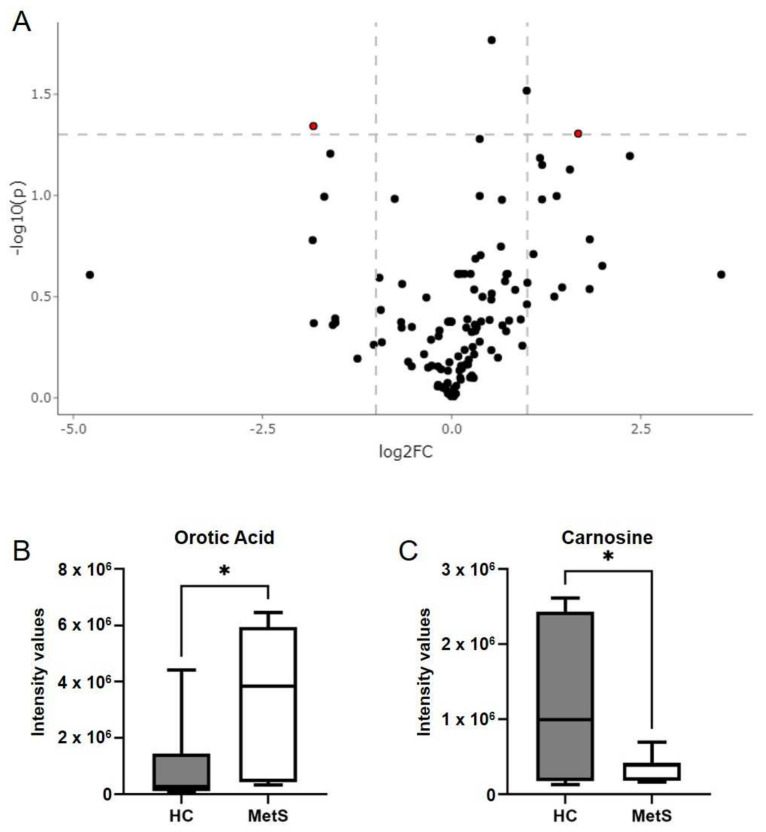
**Hybrid metabolomics of stool samples.** (**A**) Volcano plot from hybrid LCMS assays of stools from MetS and HC subjects (*n* = 7–10/group) depicting a standard panel of approx. 150 polar metabolites. Each dot represents one metabolite, dashed lines indicate default thresholds for significance (*p* < 0.05) and fold change up- or down-regulation by 2-fold ([Log2FC] = 1). The red dot on the right represents a metabolite with higher levels in MetS participants, while the dot on the left is a metabolite with lower levels in MetS in respect to HCs. Plot showing the intensity values of fecal (**B**) orotic acid and (**C**) carnosine in MetS and HC participants. Plots indicate median (±minimum and maximum). * *p* < 0.05. Two-tailed unpaired Student’s *t*-tests.

**Table 1 metabolites-12-00431-t001:** Demographics of study cohort.

Demographics		Controls	Metabolic Syndrome
**Gender**	Male	4	3
	Female	6	5
**Age**	Median	42.50	50.50
	Minimum	31	45
	Maximum	56	58
**Race/Ethnicity**	Non-Hispanic White	4	2
	Hispanic	6	4
	Native American		1
	Black		1

**Table 2 metabolites-12-00431-t002:** Putative LMSD ID of lipids with the highest fold change in the MetS group.

**Feature ID**	**Observed *m*/*z***	**Log2 Fold Change**	** *p* ** **-Value**	**Putative ID *** **(Category)**	**Main Class (Abbrev. Chains)**
5617	771.5399	4.076	0.016	Glycerolipids	Triradylglycerols (TG 12:0_12:0_22:3)
1432	558.4388	3.477	0.033	Glycerophospholipids	Oxid. glycerophospholipids (LPC 0:0/20:4;O)
4631	665.7446	3.367	0.025	Glycerolipids	Diradylglycerols (DG 19:0_20:0)
4799	680.7542	3.333	0.032	Sphingolipids	Ceramides (Cer 18:1;O3/24:0;O)
6916	989.5998	3.326	0.030	Glycerolipids	Triradylglycerols (TG 19:1_22:6_22:6)
4688	672.6672	3.326	0.042	Glycerophospholipids	Glycerophosphoethanolamines (PE P-16:0/16:1)
3675	571.3263	3.261	0.039	Fatty Acyls	Diradylglycerols (DG 13:0_20:5)
5044	700.6979	3.252	0.022	Glycerophospholipids	Glycerophosphocholines (PC P-16:0/15:1)
5270	724.7805	3.244	0.025	Sphingolipids	Ceramides (Cer 18:1;O3/26:0;O2)
5266	724.4458	3.231	0.007	Glycerophospholipids	Glycerophosphocholines (PC 12:0_20:4)
6454	905.5635	3.216	0.035	Glycerolipids	Triradylglycerols (TG 18:3_18:3_20:0)
5128	709.7706	3.201	0.029	Glycerolipids	Diradylglycerols (DG 21:0_22:6)
5129	710.1051	3.158	0.034	Polyketides	Flavonoids
7315	1371.8158	3.127	0.007	Sphingolipids	Neutral glycosphingolipids (Hex(3)-HexNAc-Fuc-Cer 34:1;O2)
4442	651.0691	3.104	0.033	Polyketides	Flavonoids
1490	531.4196	3.103	0.015	Fatty Acyls	Fatty esters (FA 36:2)
5383	739.1213	3.094	0.026	Polyketides	Flavonoids
4961	695.0953	3.077	0.032	Polyketides	Flavonoids
4980	695.7639	3.059	0.033	Fatty Acyls	Fatty amides
4982	696.0981	3.045	0.033	Polyketides	Flavonoids

* Putative ID derived from LIPID MAPS® Structure Database (LMSD) utilizing observed *m*/*z*.

## Data Availability

The datasets generated and analyzed during this study will be made available upon request.

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
