# Peer review of "Individuals with Metabolic Syndrome Show Altered Fecal Lipidomic Profiles with No Signs of Intestinal Inflammation or Increased Intestinal Permeability"

_metabolites, 2022, doi:10.3390/metabo12050431_

Round 1
Reviewer 1 Report
Comments to the author:
The aim of the study was to evaluate intestinal health in human subjects with metabolic syndrome (MetS). It is an interesting work, though carried out on a small number of participants (10 people in the control group and 8 people in the MetS group).
Major comment:
The biggest reservations are related to the selection of participants for the control group.
In section 4.1. Participants, Line 321-322 the authors stated: Inclusion criteria for the control group consisted of individuals aged 30-60 years that did not have MetS. In lines 126-127 the authors wrote: our MetS cohort showed significant differences in abdominal obesity, dyslipidemia, and insulin resistance.
The fact that the two groups differed significantly in some biochemical parameters does not mean that the control group participants were healthy. The results indicate that many of these individuals may have had 1 or even 2 MetS components. This selection of participants for the control group may have an impact on some of the results obtained.
Minor comments:
- ABSTRACT
In the abstract, the authors mention hypertension as one of the risk factors for the metabolic syndrome. This wording is imprecise. MetS definitions mention elevated blood pressure.
The goal is usually given at the end of the Background section, while the authors gave the goal in the Methods section.
- TEXT
Study design is described twice:
2.1. Study design and goal of the study
4.2. STUDY DESIGN
- METHODS
In section 4.1. Participants - the survey questionnaire is listed: ….ongoing alcohol or substance abuse using AUDIT questionnaire to determine whether the participant's behaviors were suggestive of alcohol abuse….
Please describe this questionnaire in more detail, possibly provide the source.
In section 4.2. The authors write: …. Participants' height, weight, waist, waist-to-hip ratio, and body composition via bioelectrical impedance were recorded…. Please include this data. Only fat masses and lean masses in kg are listed in Supplementary Materials.
- Limitations
There is no description for limitations of the study.
- Conclusions
While the conclusion section is not mandatory according to the journal's guidelines, I do suggest that it be added to the text of the manuscript.
- Supplementary Materials
Table 4 in the supplementary file is hardly readable. I suggest you reduce the number of digits after the decimal point.
- References
Please review critically the cited references. More than half of the cited works are published earlier than 5 years ago.
Author Response
Major comment:
The biggest reservations are related to the selection of participants for the control group.
In section 4.1. Participants, Line 321-322 the authors stated: Inclusion criteria for the control group consisted of individuals aged 30-60 years that did not have MetS. In lines 126-127 the authors wrote: our MetS cohort showed significant differences in abdominal obesity, dyslipidemia, and insulin resistance.
The fact that the two groups differed significantly in some biochemical parameters does not mean that the control group participants were healthy. The results indicate that many of these individuals may have had 1 or even 2 MetS components. This selection of participants for the control group may have an impact on some of the results obtained.
- The objective of this pilot study was to examine gut health in individuals with MetS in comparison to individuals without MetS. All participants were volunteers and put into their respective groups after clinical and biochemical assessment – thus we let the clinical data determine if they did or did not have MetS. Our criteria for MetS as we state in our manuscript is defined as an individual that has 3 of the 5 risk factors. Thus, our controls did not fit this criteria. Having MetS compared to only elevated blood pressure puts an individual at a greater risk for CVD, T2D etc. A major difference between both groups was dyslipidemia (elevated triglycerides and decreased HDL) and insulin resistances. Only one control had increased triglycerides but shown no other signs of MetS risk factors. Nevertheless, we have stated this in our new “Limitation of the study” section and agree future studies can increase the participants and to ensure all controls have no risk factors.
Minor comments:
- ABSTRACT
In the abstract, the authors mention hypertension as one of the risk factors for the metabolic syndrome. This wording is imprecise. MetS definitions mention elevated blood pressure.
– We have changed the word hypertension to elevated blood pressure in both the abstract and Introduction
The goal is usually given at the end of the Background section, while the authors gave the goal in the Methods section.
- The Reviewer is correct and we have added the goal statement to the end of the Background section.
TEXT
Study design is described twice:
2.1. Study design and goal of the study
4.2. STUDY DESIGN
- We have changed 2.1 title to “Goal of study” and 4.2 “Clinical Visit”
METHODS
In section 4.1. Participants - the survey questionnaire is listed: ….ongoing alcohol or substance abuse using AUDIT questionnaire to determine whether the participant's behaviors were suggestive of alcohol abuse….
Please describe this questionnaire in more detail, possibly provide the source.
– We apologize for the vagueness and now have cited the source of the AUDIT questionnaire and added some detail of the AUDIT questionnaire.
In section 4.2. The authors write: …. Participants' height, weight, waist, waist-to-hip ratio, and body composition via bioelectrical impedance were recorded…. Please include this data. Only fat masses and lean masses in kg are listed in Supplementary Materials.
- We have added the requested data in Supplemental Figure 1 and referenced this data in the paper.
Limitations
There is no description for limitations of the study.
– We have now added a Limitation of the study (section 5).
Conclusions
While the conclusion section is not mandatory according to the journal's guidelines, I do suggest that it be added to the text of the manuscript.
– We believe our last paragraph in the discussion section (original line 293 – 311) is our Conclusion statement and have added the title Conclusions before this section making it section 4 and the paragraph start at line 295 and ending at 313. The Methods section has now been renamed section 6 (due to the inclusion of a “Limitation of the study” section 5)
Supplementary Materials
Table 4 in the supplementary file is hardly readable. I suggest you reduce the number of digits after the decimal point.
- We thank the reviewer for making the suggestion and we have modified all tables to have 3 digits after the decimal.
References
Please review critically the cited references. More than half of the cited works are published earlier than 5 years ago.
- We agree with the Reviewer that some of the references cited are more than 5 years ago and we have added more recent references. We want to keep the older cited work as we feel they are landmark findings in the field that laid the foundation to more recent publications.
Reviewer 2 Report
The study by Coleman and colleagues aimed to evaluate intestinal health in human subjects with metabolic syndrome (MetS). I have some comments as follows:
1. Supplementary Table 1 and Supplementary Table 2.
The statistical method used to compare the two groups was not described. Please describe the statistical method in the statistical analysis section and tables. In addition, the variable such as triglycerides is usually supposed to be not normally distributed. However, the characteristics of the TG were shown as mean and standard deviation, not median and interquartile range. What were the results of the Kolmogorov–Smirnov test? If the TG variable was not normally distributed, it was shown as median and interquartile range.
2. Supplementary Table 1 and Supplementary Table 2.
In the tables, most results are given with two, even four decimals. As the precision of the measurement has only one decimal, it makes no sense to show the results with two, three, or four decimals.
3. Limitations
I suggest that the authors present the limitations of this study.
Author Response
- Supplementary Table 1 and Supplementary Table 2.
The statistical method used to compare the two groups was not described. Please describe the statistical method in the statistical analysis section and tables. In addition, the variable such as triglycerides is usually supposed to be not normally distributed. However, the characteristics of the TG were shown as mean and standard deviation, not median and interquartile range. What were the results of the Kolmogorov–Smirnov test? If the TG variable was not normally distributed, it was shown as median and interquartile range.
- We thank the reviewer for pointing out this discrepancy and now have added the statistical method that we used to compare the two groups. However, we performed the Shapiro-Wilk test instead of the Kolmogorov-Smirnov test due to our sample size. The Shapiro-Wilk test is more sensitive to non-normality than Kolmogorov-Smirnov when examining a smaller sample size. The minimum sample size for Shapiro-Wilk test is n=3. The reviewer is correct the TG are not normally distributed (Shapiro-Wilk test P=0.8503). Given normality was not significant, the unpaired Student’s t test was used. Additionally, supplementary Tables 1 and 2 have now been modified to show median and IQR and this information has been added to the table.
- Supplementary Table 1 and Supplementary Table 2.
In the tables, most results are given with two, even four decimals. As the precision of the measurement has only one decimal, it makes no sense to show the results with two, three, or four decimals.
- We thank the reviewer for making the suggestion and we have modified all tables to have 3 digits after the decimal for consistency.
- Limitations
I suggest that the authors present the limitations of this study.
– We have now added a Limitation of the study (section 5)